# Peer review of "Characteristics of a Novel ATP2B3 K416_F418delinsN Mutation in a Classical Aldosterone-Producing Adenoma"

_cancers, 2021, doi:10.3390/cancers13184729_

Round 1

Reviewer 1 Report

This manuscript described a novel somatic mutation of the ATP2B3 gene in a patient with an aldosterone-producing adenoma that harbored bilateral nodules but aldosterone production localized in one side.

Minor points.

The sentence in line 42 needs rewording. There are some reports of adrenals from patients with bilateral idiopathic aldosteronism that have been studied, they are few, but they are there with important information.

The definition of biochemical success is not according to the PASO consensus as the ARR remained very high with a marked suppression of PRA and an elevated aldosterone level. According to this criteria he had partial biochemical success as only the hypokalemia was corrected. From this point of view the patient had a mutation in the resected adrenal, but the contralateral adrenal was abnormal and most likely he had bilateral asymmetric disease.

Author Response

This manuscript described a novel somatic mutation of the ATP2B3 gene in a patient with an aldosterone-producing adenoma that harbored bilateral nodules but aldosterone production localized in one side.

Minor points.

The sentence in line 42 needs rewording. There are some reports of adrenals from patients with bilateral idiopathic aldosteronism that have been studied, they are few, but they are there with important information.

Response:

Thank you for comment.

We have reworded the indicated sentence as follows:

Primary aldosteronism (PA) is originally classified into unilateral hyperaldosteronism and bilateral hyperaldosteronism (BHA). BHA is mainly related to bilateral idiopathic hyperplasia which could not be detected by computed tomography (CT). Bilateral aldosterone producing adenoma (APA) could be detected as bilateral detectable mass by computed tomography (CT) but is a rare finding. (Revised manuscript line 43 to 47)

The definition of biochemical success is not according to the PASO consensus as the ARR remained very high with a marked suppression of PRA and an elevated aldosterone level. According to this criteria he had partial biochemical success as only the hypokalemia was corrected. From this point of view the patient had a mutation in the resected adrenal, but the contralateral adrenal was abnormal and most likely he had bilateral asymmetric disease.

Response:

Thank you for comment. We have revised the relevant text to describe that his PASO biochemical results reached partial success and the contralateral adrenal gland could also be abnormal. (Revised manuscript line 31, line 169 to 171 and line 268 to 284).

Reviewer 2 Report

The authors reported a novel ATP2B3K416-F418delinsN mutation in a patient with unilateral PA who underwent adrenalectomy and present its clinical features. Although several genes causing aldosterone-producing adenomas have been reported in recent years, there are still many unknowns, and the reported gene is an important piece of the puzzle.

I propose the following points to make their worthy reports more solid scientific evidence.

1. It should be confirmed and noted that genes such as KCNJ5 and ATP1A1, which are somatic mutated genes that promote the secretion of adrenocortical hormones, are not expressed in this case. The patient may have multiple gene mutations that cause the condition. In such cases, it cannot be said that the reported the new mutation was dependent as the cause of the typical clinical picture of PA as in this case.

2. In addition to evaluating the WBC and adenoma mutations, additional evaluation of the genetic mutations in the adrenal tissue adjacent to the adenoma would be helpful to make a stronger case for the adenoma origin.

3. The new gene mutations reported in this study are assumed to be infrequent causes of PA. It would be more important to describe the clinical implications and applications of the newly discovered infrequent causative gene.

Author Response

  1. It should be confirmed and noted that genes such as KCNJ5 and ATP1A1, which are somatic mutated genes that promote the secretion of adrenocortical hormones, are not expressed in this case. The patient may have multiple gene mutations that cause the condition. In such cases, it cannot be said that the reported the new mutation was dependent as the cause of the typical clinical picture of PA as in this case.

Response:

Thank you for comment.

We have used Sanger sequencing to confirm that there were no other conventional and well-characterized aldosterone-driving gene mutations, including KCNJ5, ATP1A1CACNA1D, and CTNNB1, in the adenoma harboring with ATP2B3 K416_F418delinsN mutation (Supplementary Figure 1). The primer sequences and detectable mutation sites for the KCNJ5, CACNA1D, ATP1A1, and CTNNB1 genes are shown in Supplementary Table 1.

We have emphasized this in the end of the first paragraph of the Discussion in order to relate the functional APA to this novel ATP2B3 mutation. (Revised manuscript line 238 to line 241). 

Supplementary Figure 1 Sanger sequencing analysis of tumor DNAs for conventional and well-characterized aldosterone-driving gene mutations, including KCNJ5, ATP1A1CACNA1D, and CTNNB1, in the adenoma harboring with ATP2B3 K416_F418delinsN mutation. (Figure S1, please see attached file or revised manuscript)

Supplementary Table 1. Primers used for Sanger sequencing

ATP2B3 (NM_001001344.2)

Detectable mutations

Base pair

ATP2B3-Exon 8 (1)

Forward

CCTGGGCTGTTTATCCTGAA

p.Val424_Leu425del

p.Leu425_Val426del p.Val426_Val427del

416

Reverse

CCCCAGTTTCCGAGTCTGTA

CACNA1D (NM_000720.4)

Detectable mutations

Base pair

CACNA1D-Exon 8 (2)

Forward

GCCTTGATGACTCTGTGTG

p.Gly403Arg

p.Gly403Asp

453

Reverse

CCAGCAAAGCTTGTGTGGT

CACNA1D (NM_001128839.3)

Detectable mutations

Base pair

CACNA1D-Exon 16 (2)

Forward

TTTACTTCTGTAGACTGTCC TTTTA

p.Phe747Leu

p.Ile750Met

367

Reverse

ACACGTGACTCCCACTCTCAGC

CTNNB1 (NM_001904.4)

Detectable mutations

Base pair

CTNNB1-Exon 3 (3)

Forward

CATTCTGCTTTTCTTGGCTGTC

p.Ser33Cys

p.Gly34Arg

p.Ser45Phe

p.Ser45Pro

p.Ser45Cys

483

Reverse

GCTATTACTCTCTTTTCTTCCC

ATP1A1 (NM_001160233.2)

Detectable mutations

Base pair

ATP1A1-Exon 4 (1)

Forward

TTCCTTGGGCCTATTGTTTG

p.Gly99 Arg

p.Leu104Arg

487

Reverse

GTGGGAGACAAAGACGGAGA

  1. In addition to evaluating the WBC and adenoma mutations, additional evaluation of the genetic mutations in the adrenal tissue adjacent to the adenoma would be helpful to make a stronger case for the adenoma origin.

Response:

We have confirmed that the adrenal tissue adjacent to the adenoma was carrying wild-type ATP2B3 gene by using Sanger sequencing (showed in the revised Figure 1A); added to the end of Discussion 4.2. (Revised manuscript line 265 to 267).

Figure 1. A novel ATP2B3 K416-F418delinsN mutation in classical APA. (A) The deletion at nucleotide 1248 to 1253 as GTTCTT in the ATP2B3 K416-F418delinsN gene was identified in a patient with APA in the resected adrenal adenoma. The amino acid residue 416 to 418 of ATP2B3 protein was substituted from lysine (Lys) and 2 phenylalanine (Phe) to asparagine (Asn) insertion. The letters for nucleotide bases represented as following: C, cytosine; G, guanine; T, thymine, (B) the protein secondary structure of mutant ATP2B3 channel was demonstrated. The yellow circle N represented insertion of amino acid, asparagine. The model was based on Protter software application (http://wlab.ethz.ch/protter/start/). (Revised Figure 1, please see attached file or revised manuscript)

  1. The new gene mutations reported in this study are assumed to be infrequent causes of PA. It would be more important to describe the clinical implications and applications of the newly discovered infrequent causative gene.

Response:

Thank you for comment. We have described the clinical implications as follows:

Different mutant genes of PA have specific pathophysiological, clinical and biochemical manifestations. Identifying pathogenic genes in PA would help physicians determine course of the disease and make decision on treatment and follow-up. Patients with ATP2B3 mutant APA have obvious aldosterone and potassium abnormalities. Most reported ATP2B3 mutant PA were unilateral APA. Our finding of this new mutation will help researchers in this field to incorporate this mutation in their future routine screening of the possible mutation spots, and the actual prevalence of this novel mutation will be further assured. (Revised manuscript line 285 to 295).

Round 2

Reviewer 2 Report

This second version of the paper is a great improvement, the authors are to be commended.